# Ethics and Care: For Animals, Not Just Mammals

**DOI:** 10.3390/ani9121018

**Published:** 2019-11-22

**Authors:** Jennifer A. Mather

**Affiliations:** Department of Psychology, University of Lethbridge, Lethbridge, AB T1K 3M4, Canada; mather@uleth.ca

**Keywords:** ethics, invertebrates, fish, welfare, fair evaluation

## Abstract

**Simple Summary:**

Animals come in a huge array of types, species and structures, but without realizing it, we have focused our care on those like us, mammals. They look something like us, so it’s easy to empathize with animals like dogs, cats and horses. We see them on magazine covers, worry about their pain and suffering and try to conserve their habitat. Because of this tight focus, we lose all the other animals—invertebrates, for instance, are 98% of the animals on the planet. Even though these animals aren’t mammals, we should still care about their welfare. This paper gives the example of two groups, fish and crustaceans, whose welfare has been overlooked or denied, as an example of how we should care about the well-being of all animals, not just mammals.

**Abstract:**

In the last few decades, we have made great strides in recognizing ethics and providing care for animals, but the focus has been mainly on mammals. This stems from a bias of attention not only in research but predominantly in non-scientists’ attention (to ‘popular’ animals), resulting partly from discussion about and depiction of animals in publications addressed to the public. This is somewhat due to political pressure, and can result in uneven conservation efforts and biases in targets for welfare concerns. As a result, there has been a huge backlash again, with concerns about pain sensitivity and welfare in fish, and a less focused but more pervasive omission of consideration of all invertebrates. That means welfare efforts are focused on 0.2% of the animal species on the planet, and education about non-mammals, particularly addressed to children, is necessary to broaden this focus and care more fully for the inhabitants of the planet.


*“All animals are equal, but some animals are more equal than others”.*
—George Orwell

## 1. Introduction

It is less than 50 years since scientists realized that some species of animal deserve moral consideration from us, especially true for species similar to us such as monkeys [1]. However, when we view, report and care for animals, we are not objective in our liking. There is a huge bias in favor of mammals, which is not consistent with their frequency (5400 mammal species or 0.2% of those on the planet, versus about 20,000 fish and at least a whopping 900,000 insects). Instead, we care about animals that are similar to us [2], and invertebrates especially are neglected [3]. As well, there is often a huge disconnect between the science about animals’ welfare and the procedures carried out by those who actually use these animals [4]. These considerations, particularly obvious when discussing fish welfare [5], are the result of a chain of interactions between science and ethics, also fueled by public opinion and politics. There is an interacting triumvirate of lack of public knowledge of the scope of animals, little attention to non-mammals by policy makers and minimal scientific research devoted to them [3]. Everywhere we hear about mammals positively and predominantly and extend care to them, our research increases our understanding of them and public opinion influences the steps we take to care for them [4,6,7,8]. Almost no one cares about treatment of ‘lower’ animals [3] so we mostly do not care for or about non-mammals. But we must.

## 2. Why Do We Have This Bias?

Humans, in general, are anthropomorphic, most concerned with us and related species. When deciding which animals should be given moral consideration, Broom [2] and Jones [4] suggest that humans base our judgment on several characteristics: their cognitive complexity, wide learning ability, large brain size, indications of pain, and awareness. Only animals similar to humans are assumed to have these four capacities [9]. Note that the list of animal species selected for their study shows their biased attention to them, as it consisted of 10 mammals, 6 birds, 3 reptiles, 2 amphibians and fishes and only 3 animals representing all the invertebrates. Familiar behavior that we see in mammals such as our pet dogs leads to human attribution of these mental abilities and states, eventually forming the groundwork for empathy and concern for welfare. This concern also extends to what can be called interpretive anthropomorphism [10], attribution of intentions, emotions and beliefs to non-human agents. A case emphasized by fish biologists is Disney’s Finding Nemo. Anemone fish are protandrous hermaphrodites, which means that a family inhabiting an anemone is led by an adult female, secondarily by an adult male. If the female is lost (as in the movie), the male changes sex and becomes the dominant female, and the oldest immature (in this case, Nemo) becomes the male. Needless to say, this did not happen to Nemo in the movie; instead, they presented his emotions and responses as those of a human adolescent. If animals are not similar to us, we may pretend that they are, thus limiting our understanding of potential behavior and ecology, which must guide how we manage their care.

When animals have similar motor actions and facial expressions to us, we empathize with them, and so we ‘like’ and care for them. Individual values of other animals depended more specifically on prior relationship to humans, perception of values such as attractiveness and intelligence, the individual’s previous experience with them and his or her attitudes to nature and wildlife [11]. Given this, her subjects in the UK clearly preferred animals similar to us. Such a determination of which are ‘important’ species [10,11] affects many human activities, including conservation efforts and funding, focus of scientific research and even efforts at reintroduction and habitat conservation, impacting their welfare and reflecting back on the very survival of any species. Because humans are responsible for the loss of biodiversity, Montgomery [12] points out that we must understand these values humans place on animals so that we can reverse it. But this ’value’ is not based on species number or ecological importance but on our social cognition and the way in which we understand and position them, and the American general public gives animals a ‘social construction’ [13] in the sense that they are viewed subjectively by them. This leads to ‘political power’, as different species are valued not just directly but also by the formation and pressure of interest groups (NGOs). In their study, mammals and birds were ranked highly in ‘mattering’, followed somewhat lower for fish, and these taxa were rated much more highly than other animal groups. This positive viewpoint and public power, particularly obvious for birds (think of the Audubon Society and Ducks Unlimited), correlates strongly with benefits given by the US Endangered Species Act. In other words, how the public constructs the importance and attractiveness of species groups is a direct link to how we behave towards them, and thus their care and even survival. Conversely, if we construct types of animals as food [14] or as a threat [15], we do not care about their welfare and conservation.

To direct care equitably, we must understand how the public feels about animals in order to properly care for them. In general, there are two types of motivational determinants of attitudes we have towards animals [16], *affect* or feeling, and *utility* or usefulness. The affective dimension is a powerful motivator, as if we can be brought to care about any animal group or species, we will act ethically towards them. Yet the amount of knowledge about sharks, used as a model group, correlated with the positive attitudes to them [17], so knowledge and feelings can be combined. Still, when Australians were asked to select animal species that would be saved on a hypothetical Ark, chosen animals were those similar to humans and predominantly mammals [18]. Since their sample was also biased, taken only from mammals, birds and reptiles, we see that these groups defined animals for the researchers as well.

## 3. Attention Guides Consideration: See Popular and Unpopular Animals

What animal species do people ‘like’? These are mostly what has been called ‘charismatic megafauna’, again predominantly mammals. Entries for North American wild animals on Twitter found polar bears overwhelmingly mentioned, followed by bison, brown bears, cougars and orca killer whales [19]. Large mammalian predators are seen as charismatic [20], and huge amounts of money are then spent on their care and conservation. While these have historically been Flagship species featured by zoos and aquaria [21], further assessment suggested that the Flagship category could be extended with more education. In fact, these few species have been over-emphasized so that less ‘obvious’ animals have a novelty effect. Given this narrow focus, the ‘big five’ animals sought by tourist visitors to sub-Saharan Africa—elephants, giraffes, lions, leopards and rhinos—have been seen as conservation tools, in that protecting them fosters conservation of important ecosystems of which they are a part [8]. This is unfortunately only partly true, as they are ‘umbrella’ species for some but not all diverse ecosystems. As well, increased marketing of ‘unappealing’ species could increase donor behavior to animals in campaigns by the World Wildlife Fund and the Zoological Society of London [22], possibly because the charismatic species have already been so exploited.

As one goes ‘down the phylogenetic tree’, vertebrates appear less and less like humans and thus their appeal wanes. This is particularly true of fishes, lacking the faces that hold the expressions that we see as indicating positive or negative reactions. Furthermore, as Brown [23] points out, with the exception of snorkelers and scuba divers, we rarely see fish alive in their natural environment. Fishes are not seen by the general public as intelligent, despite the fact that many are. Remember, human expression of concern for species conservation correlates with the charisma expressed for the mammals as above, and not with their endangered status [6]. This anthropocentric view is central to assessment of fishes, which are widely caught as food [24], cultivated in aquaculture [25], used as models in research [23] and as ‘ornamentals’ [26]. Equally, with more species than are found in all other vertebrates combined, fish are going to vary widely in capacities such as learning, cognition and ‘emotion’ [27].

If animals *really* do not look like us, the public generally does not care about their welfare. In a pioneering study of perception of invertebrates, Kellert [28] asked people to rank 33 species of invertebrates in importance and explain their attitudes towards them. He pointed out that the predominant affective attitudes to them were fear and disgust, occasionally positive towards attractive species such as butterflies or utilitarian ones to food species such as shrimp. Public knowledge about them was minimal; as for instance only 27% knew that an octopus was not a fish. Remember that if we do not care *about* an animal species or group, we will not care *for* it. Ranking within the arthropods by children showed that the most positive ones were for butterflies, the most negative to earwigs [29]. This negative viewpoint of these ‘lower’ animal starts in childhood, as Swiss children showed disdain for land invertebrates [30], less so for those that fly than animals that creep. The bias starts as early as kindergarten, when Italian 3–6-year-olds preferred large, mammalian and domestic animals [31]. However, the bias is again notable in the contents of the questionnaire presented to them, with 22 mammalian examples, six birds, four reptiles, two each of amphibians and fishes and 12 invertebrates. These numbers in no way reflect their relative frequency, as around 98% of all animals are invertebrates and the number each of the other vertebrate classes is at least double that of mammals. In other words, experts fuel the public’s biases, which are then fed back to us.

This negative attitude is entangled with fear of animals that are sometimes ‘risky’. Jancova et al. [32] looked at affective emotions and attitudes of Czech people to pictures of representative reptiles. Generally, subjects showed correlated emotions of fear and disgust, and addressed them particularly to snakes and legless lizards, with crocodiles ranking higher on fear. Otherwise, turtles and lizards grouped together. Snakes may have evoked an evolutionarily adaptive threat, as some are poisonous, and ethics has a long way to go with this attitude. Still, the authors point out that educational programs might be addressed to these real concerns, and education is necessary to change these attitudes. In fact, a wide-ranging study [33] found that children’s fear of snakes varied amongst countries and was thus more cultural based than genetic. Spiders are negatively stereotyped in Western but not all cultures, e.g., Navaho weavers and Anishinabee dream catchers [34]. Yet both Slovakian and South African high school student feared them [15]; few were interested in a scientific or naturalistic dimension of these animals and nearly half felt they would be nervous if a spider was near. They are not going to be concerned with their welfare. Amphibians are not dangerous, just different, and are highly endangered all over the world. Yet when Slovenian children were asked about fear and disgust, amphibians were mid-range in fear (well below scorpions, wolves and snakes), while children showed a very high level of disgust particularly to toads [35], who were most clearly seen as ugly and slimy. An ethic of care is going to have a lot of problems here. Most of the children, however, had nearly no direct experience with frogs or toads—or for that matter with the highly feared wolves and scorpions, so the biased attitudes were socially acquired, and education can reduce them. Proof of this is that Schlegel et al. [29] found that children who could recognize animal species had a more positive attitude to them.

## 4. Attention Directs Action: How Is Bias Carried Out?

Bias can occur in unexpected ways and has practical as well as attentional effects. On covers of ten US conservation and nature magazines, including well-known ones such as Audubon and Natural History, pictures of animals predominantly featured mammals at 40% and secondarily birds at 17% [36]. The species repeatedly shown were large mammalian carnivores, with the result that wolves, bears and large felids were dominant, and thus are seen as the animals we should care for and about. In a much smaller example, from 24 children’s books submitted to the Animal Behavior Society Children’s Book Competition in 2019, 16 had mammals as a subject, with a sprinkling of one per bird, reptile, amphibian and mollusk and a story on animals of the leaf litter, and three books crossing taxonomic groups. Writers are also telling children that the name of animals is mammal. From an exhaustive study to find charismatic animals, Albert, Luque and Courchamp [20] generated an online survey, asked primary students from Western countries, evaluated zoo websites and checked publicity for Disney and Pixar movies. Of the ‘top 20’, 19 were large, 18 mammals and 17 terrestrial. With four big cats, three bears and a canid, exotic large predators predominated. To sum it up, what do the public say represents an animal? Large land mammals.

Biases in favour of land animals, as seen above, neglect the marine environment, which constitutes 70% of the area of the planet. It is little featured in communication to the public [14], but even that emphasizes mammals again. There are about 130 species of marine mammal and over 20,000 fishes, but dolphins and whales are the only marine species mentioned as charismatic [20]. They were given positive, aesthetic and humanistic viewpoints although, ironically, with a utilitarian bias and little real understanding of their biology [37]. As for land mammals, can protecting top predators lead to preservation of the ocean ecosystems they live in [38]? Unfortunately, protection often extends to very small areas with extreme concentration of one species, as in the breeding grounds of sea lions or puffins, but this may not be enough. Scammon’s Lagoon is protected for grey whales but they only breed there, and must migrate to Alaska to feed, hopefully surviving along the way and finding enough food there. The northern right whales are colliding with ships in the Gulf of St Lawrence with deadly consequences, so a speed limit has been put into effect. In contrast, Finding Nemo led to many anemone fish being captured and kept in home aquaria, but, sadly, no one enforced a regulation that they must be kept in their natural home of living anemones. In general, tourism is less obvious for marine than terrestrial mammals, as failing fisheries pinpoint problems yet charismatic species still receive the most attention [38]. In the US Pacific northwest and the Canadian BC coast, orca killer whales and salmon are both endangered, the first because of the second. But publicity for fish conservation is focused on how we can have enough fish to catch, for orcas on how these particular pods (in general, orcas are thriving) can be preserved. Ethical consideration of animal confinement, emphasized by the movie Free Willy, led to a drastic reduction in the presence of orcas in aquariums, but no one has done the same from equally confined but mobile sea turtles.

These attitudes lead to a bias towards mammals in conservation programs. For targets of projects sponsored by the San Diego Zoo and the Wildlife Conservation Society [39], there was such a bias against amphibians when evaluating conservation programs under the US Endangered Species Act (ESA). Despite their world-wide ecological threat, amphibians received the least funding of all the vertebrate classes, and the ESA also under-represented the number of species designated as threatened or endangered. In the UK, the EU Life Nature program gave 19% of its funding to projects on mammals (0.2% of animals) and 41% to birds (0.7%) but only 8% to arthropods (82%). We do not ‘like’ them, we do not care about them [7]. This can also result in a differential supply of money for their care; a recent fundraiser for the Vancouver Aquarium highlighted the possibility to ‘adopt’ a marine animal, but listed only sea otters, sea lions and sharks.

Scientific journals that study or report on animals should represent them equitably. An informal survey of the representation of animals of different groups by reading the titles in the index of recent editions, eliminating those with no phylogenetic descriptor even after the Abstract was checked, did not show this. They were (1) *Journal of Applied Animal Welfare Science* (JAAWS), described on the web as covering “The science of animal welfare”; (2) *Anthrozoos,* “A multidisciplinary journal of the interactions of people and animals” (3) *Animals,* self-described as “devoted entirely to animals” (4) *Applied Animal Welfare Science* (AAWS), focusing on “Behaviour of domesticated and utilized animals” and (5) *Animal Behaviour*, a research-based journal on “research on all aspects of animal behavior”. The percentage coverage of different animal groups varied but it was always heavily weighted towards mammals (Table 1). Papers in JAAWS and *Anthrozoos* focused nearly totally on mammals, at 93% of their articles. If you read *Anthrozoos*, you have the impression that we only interact with mammals, and if your read JAAWS that we only care about their welfare. *Animals* and AAWS were not quite so heavily weighted, at 74% and 80%, with 19% and 15% of their papers on birds. JAAWS and AAWS had no recent papers whatsoever on invertebrates and only a few on ‘lower’ vertebrates, *Anthrozoos* and *Animals* were only slightly better. No wonder the public does not care about ‘lower’ animals—they do not exist for them.

The subject groups of recent papers in *Animal Behaviour* were much more diverse, with 32% mammals, 21% birds and a similar 21% on insects. This looks like a more representative diversity until one remembers that there are twice as many bird as mammal species and 170 times as many insects. Non-insect invertebrates were hugely under-represented, with 2% spiders, 3% crustaceans, and a single paper each on gastropod and cephalopod molluscs, the whole phyla Echinodermata and Annelida. For scientific journals, animals are still mainly represented by mammals. Are scientists not free to consider species that generate interesting problems for them, rather than proportional representation? Of course, but if we do not look widely, we will not see ‘interesting problems’ that fall outside the scope of mammalian diversity. Think of the cephalopods, only recently featured for their different route to intelligence and cognitive ability [40,41] and equally recently the candidate for ethical concern in the European Union by 2010 [42]. Part of the problem of lack of care for non-mammals is that we do not know much about them. We will not know about them if we do not study them, and scientists are not studying them.

## 5. What Does This Lack of Attention Mean for Welfare of ‘Overlooked’ Animals?

### 5.1. Fish Pain and Welfare

The most contentious research area in non-mammalian welfare is the debate on whether fish feel pain, with opposing sides almost trading insults in *Animal Sentience*, after Kay’s [43] feature article on why fish do not feel pain. Lost in the background are some of the basic assumptions about animal welfare, its definitions and its biological bases. Broom [2] pointed out that opioid systems that regulate pain are extremely widespread across the animal kingdom, and evidence of their physiology, including the opposition of the antagonist naloxone, has been proven not only in mammals, but also in several insect and crustacean species and even in gastropod mollusks. It is logical that effective neurotransmitter systems should be widely phylogenetically preserved, but important to realize that the physiology of this system is pretty well universal. Of course, reception of damaging or potentially damaging signals does not feed subjective reality in animals with simple nervous systems; this is considered ‘only’ nociception. Jones [4], in an excellent and wide-ranging review of animal sentience and welfare, quotes the definition of pain in humans, adopted by the International Association for the Study of Pain, as “an unpleasant sensory and emotional experience associated with actual or potential damage, or described in terms of such damage”. While it can be criticized for its emphasis on subjective experience, it is very clearly a central and not a peripheral phenomenon. Thus, to have pain, an animal must have some kind of sentience, and clearly pain is a product of brains.

This is where the problem and the debate come in. Animals that feel pain must be agreed to have sentience, and Broom [2] pointed out that this involves a complex life/behavior, widespread learning, a complex brain, and awareness. Brown [23], probably the world expert on fish behavior, presented much evidence in his paper in the scholarly journal *Animal Cognition* that fish indeed have these characteristics. He also noted that present-day fish species are not ‘primitive’. They are the oldest vertebrate group, but “have not been standing in an evolutionary backwater all that time” (p 2). His information is supported by that of Braithwaite [27] who points out that fish have parallel brain development to that of mammals. Her small book, Do Fish Feel Pain [44], is an elegant and simple presentation of evidence. She uses Dawkins’ [45] criteria for welfare, that the animal is 1) in good health and particularly 2) that it will seek out situations that it ‘wants’.

Countering this information about behavior, Rose [46] and Kay [43], both neuroscientists, suggest that pain as defined is a product of the mammalian brain, particularly cortical areas of it. Since pain is a product of mammalian brains, they argue, it cannot be defined as pain if it is experienced by a non-mammal. This is an extreme example of mammal-centrism, based solely on neural structure. They cast doubt on the experimental procedures of Braithwaite et al. [27]; when different points of view are advocated, it seems the research is subject to unusual and excessively rigorous scrutiny. They explain that we know little about all the different species of fishes, echoing Brown [23] but with a different point of view. Birch [47] suggested a precautionary principle for welfare consideration, the idea that if we do not actually have information on precisely that species and situation, we should still advance welfare protection to animals. These researchers deride that principle, calling it non-scientific, yet in the absence of exact information about the specific species or even sex or developmental stage, it is clear that some compassion should be demonstrated. Given our vast areas of ignorance about non-mammals, this seems logical. For instance, the AZA guide to the welfare of the benthic *Enteroctopus dofleini* can easily be applied outside the genus to the similar *Octopus vulgaris* when the European Union regulations on care of cephalopods are enforced, but would be hopelessly inapplicable to the pelagic dimorphic drifter *Argonauta argo*.

#### Why Such a Heated Debate?

Fishes have not been accorded protection by the US Animal Welfare Act of 2010, which protects only mammals and not all of those. Fish are a major food group, around 183 million metric tonnes were ‘harvested’ in 2015. Humans exploit fish in several situations including wild harvesting, amateur catch-and release fisheries, ornamental fish possession by hobbyists, and aquaculture, which is the fastest expanding use of these animals. Wild harvesting of fish has a number of ethical challenges [24] including long periods of ‘suffering’ between capture and death, damage after escape, and poor survival of captured ‘by-catch’ particularly by netting methods. The ornamental fish trade has been almost unregulated [26] and major welfare health issues include high mortality and stress from capture and transport. The aquaculture industry had come under scrutiny [5] for ethical issues of keeping animals in captivity, including poor environmental conditions, crowding and stress (there are also ecological issues about damaging the environment in areas where these fish ‘farms’ are set up). Most ethically troubling is the catch-and-release recreational fishery. After all, it is for the pleasure of the humans only, and there is evidence that hooking damages the fish, stresses them, and produces changes in their behavior [48].

Perhaps it is because there are such clear consequences to providing for fish welfare in these situations that there is so much heat in the debate about whether the fish actually feel pain. If they do, all these activities will have to be regulated. Browman et al. [49] predict a kind of ‘regulatory Armageddon’ where any fish-related activity is severely regulated, curtailed or banned, although this has not happened. This includes a suggestion that the most stringent welfare regulations of fish research will be applied for publication about fish carried out across the globe into countries that do not have such regulations, choking off research. Birch [47] points out that bureaucracy does not actually choke off such necessary research. Instead, we need a lot more regulation of activities that affect any species that is not a mammal, including conservation for non-mammalian vertebrates and more attention and money for research on their welfare. We need consideration where and how it is appropriate and not caring for these animals and simply ignoring those others. As well, better fishing practices produce better ‘quality’ fish, and in Scotland these RSPCA-endorsed ‘Freedom Foods’ fish fetch higher prices. Ironically, both sides of this acrimonious debate suggest more research. More knowledge would likely lead to more and better care. Given the diversity of fishes, we should apply what we know of their welfare to other related species, as Birch’s [47] Precautionary Principle would suggest. But note the lack of publication of papers about fish, their welfare and fish-human interactions in the journals mentioned in Table 1 above. The Cooke and Sneddon [48] paper, published in *Applied Animal Behavior Science,* is a rare exception.

### 5.2. Welfare of Invertebrates

The situation of invertebrate welfare is much broader and therefore not so clear-cut, yet in some ways more pervasive than that of non-mammalian vertebrates. Regulators, with some exceptions, have not thought much about welfare protection and ethical treatment for any invertebrates, though see [49] for an up-to-date table. Five countries (Canada, New Zealand, Australia, Norway and Switzerland) and the European Union protect cephalopods in research. New Zealand, Norway and Switzerland protect decapod crustaceans, and Norway protects honeybees [50], a logical step since they are so important to crop pollination. That is not many. The US Animal Welfare Act excludes non-mammals [4], as well as rats, mice and farm animals, poor protection indeed.

Why have invertebrates been so poorly protected? Remember that for an animal to be believed to show pain it should be also deemed to be sentient, though Dawkins [45] believes it is only important to determine if an animal can suffer. Only recent research has found complex problem solving in invertebrates and raised the possibility that they have ‘minds’. Even in 2013, research showed [2] that honeybees and ants have cognitive maps, *Portia* spiders can make out-of-sight detours and that stomatopod crustaceans may use deception. More recent evidence shows that octopuses can plan an action that will be useful to them in the future and specialize in exploration ’merely’ for acquisition of such information [41], and that invertebrates from two different phyla can play. As research expands into different situations for different animals, the scope of invertebrates’ known behavior broadens. De Waal, in a conference presentation, suggested that after a set of scientists find a cognitive capacity in one animal group, others test and find similar abilities in unrelated ones. There is a bias to study the known and look for stereotyped capacities but even so, no one can pretend that invertebrates are just ‘things’ any more and ‘draw a line’ in consideration of their welfare.

The possibility that invertebrates might be sentient led the European Union, when revising its animal welfare legislation, to evaluate whether non-vertebrates were sufficiently sentient that their welfare should be of concern [2]. Cephalopod molluscs and decapod crustaceans were presented as candidates for inclusion. Eventually, decapod crustaceans were excluded because of the objection of the UK, possibly on the basis of commercial interests, suggesting that the evidence of pain did not support the claim [51], though see [52]. Why cephalopods, especially octopuses? Public information about their curiosity and flexibility—remember octopus Paul, who was supposed to predict Germany’s performance in the World Cup—as well as scientific information [41], has accumulated over the years. Though physically very different from vertebrates, they are similar in this competence. In 2012, the Cambridge Declaration on Consciousness included them in animals that might be conscious [53]. Ironically, they might have been spared the opposition we see to welfare regulation of activities involving other groups because in the Western world they are not seen as commercially important, though they are a major source of food elsewhere, and the decimation of the top ocean predators means that cephalopod populations are increasing in the oceans [54].

Inclusion of cephalopods in the Directive of the European Union (2010) was an important step in ensuring their welfare. So much was not known about cephalopods that, *contra* to what Browman et al. [49] would have predicted, an agency to investigate their welfare, called Cephs-in-Action was set up and quickly began research. They investigated stress and disease, anesthesia and euthanasia, and indices of cephalopod welfare, also forming a training program for cephalopod workers. Andrews et al. [55] covered the practical applications of the new law, Smith et al. [56] the ethical procedures and Fiorito et al. [42] a set of guidelines for their care. Butler-Streuben et al. [57] published a guide to the physiology of anesthesia and euthanasia. Having regulation does not appear to be pushing cephalopod research out of the EU to be carried on by nations with less clear welfare expectations, a concern addressed by Birch [47], who also did not believe that it would happen. Rather, some international journals require more stringent evaluations of cephalopod welfare and researchers are matching them, even though their country of origin does not require it. Note that objections were made to planned octopus ‘farming’ because it would be cruel to now possibly sentient animals [58], although this could be argued to apply to many species already farmed. Perhaps improving the welfare of all farmed animals should instead be their goal.

#### What of Decapod Crustaceans?

The scientific panel that reviewed the extension of welfare in the EU to invertebrates suggested the decapods did have pain. Strong criticisms were raised by non-researchers such as Diggles [51] and objections from the UK forced their exclusion despite suggestions that they could suffer [52]. This might again have been pressure from commercial interests and, similar to the situation for fish. Ironically, much of the research (for more information, refer [59]) was based on Elwood’s long-term studies of the motivation of hermit crabs to gain protective shells, a thorough intellectual base for these comparison despite crustaceans being poorly studied in general. The anthropocentrism of the position that crustaceans cannot feel pain, as for fish, is clear [52]. One procedure that raises particular questions is the ablation (removal) of the eyestalks of commercially raised prawns. An experimental study [60] showed that anesthetic significantly reduced tail-flicking, rubbing, leaving shelter and disorientation after the procedure. Even with anesthesia, chopping off the eyestalks to bring prawns into breeding readiness does not sound like good welfare. Because of this research, one of the largest producers of prawns in Latin America stopped using the process altogether, proof that such work can reduce the likelihood of commercial operation to do damaging procedures [52]. And an organization called Crustacean Compassion has begun to campaign in the UK for legal protection for decapods.

The situation for welfare of insects is far starker, possibly because, with few exceptions, they are perceived as nuisance or pests. Butterflies are seen as attractive [29] and bees as useful [61] and the latter therefore as an important target for welfare consideration [50]. However, Cunningham [61] points out that the public attention has resulted in some speciesism, skepticism, narrow focus and sometimes human-interest pragmatism, this last echoed by Garrido and Nanetti [50]; welfare should not be about ‘how we can get the most gain from the situation’. Cardoso et al.’s [62] ‘shortfalls’ are important here, as they note we have not even identified most invertebrate species (echoed for all the marine animals by the Census of Marine Life project), we do not know their distribution, their abundance and their natural history, i.e., ecological niche. Both these sources emphasize that with this lack, we are going to drive many species extinct before we even know they exist. Boppre and van Wright [63] point out the welfare dilemmas that ensue when we keep insects in captivity.

## 6. Conclusions

What do we do about this attitude, called ‘institutional vertebratism’ by Leather [64], when discussing invertebrate conservation? The solution is surely education, as emphasized by many of the authors mentioned in this paper. Unwittingly in many cases, science and dissemination of our discoveries has told us that the name of animal is mammal, also a result of the triad of public, political and scientific pressures [62]. The public has to know about what the famous conservationist EO Wilson called “the little things that run the world”, and this is beginning to happen as discoveries of important and interesting abilities in ‘lower’ animals, as is beginning for cephalopods in particular [40,41]. Zoos and aquariums, magazines that educate us about animals, films and media accounts have to give a more accurate picture of what constitutes ‘animals’. This is so important because the political biases in terms of what animals are protected and how and what we do to animals in our care is strongly influenced by the public. Public influence will help correct the ‘skew’ in conservation projects, focusing more funding for groups such as insects [64] and amphibians [39], and less focus on some such as marine mammals [20]; remember the bias towards charismatic species [6]. Scientists, too, have to stop focusing so much on ‘animals like us’, and the secondary journals have a special obligation. What if JAAWS or AAWS had a Special Issue each on the welfare of reptiles, amphibians, fish, insects and marine invertebrates? The contents of this paper show that there are definitely enough problems in each area to fill an issue. And in the end, it may be educating children that is the most important set of actions. Remember, children had negative attitudes to non-mammals but had no direct encounters with them [35], and the more children knew about insects, the better their attitude to them [29]. Attitudes of children to snakes were the result of culturally induced biases [33] and education could overcome these biases. We need to work in all these areas at once to make it clear that the definition of animal is not mammal, but perhaps the education of children for the future is our best opportunity.

## Figures and Tables

**Table 1 animals-09-01018-t001:** Percentage of different animal groups in recent publications of welfare-related journals.

Phylum	Class	Species	JAAWS ^1^	Anthrozoos	Animals	AAWS ^2^
			*n* = 261	*n* = 129	*n* = 526	*n* = 224
Chordata	Mammal	5400	93	93	74	80
	Bird	10,000	3	2	19	15
	Reptile	10,000	3	2	<1	0
	Amphibian	7000	<1	1	<1	
	Fish	20,000	2	1	4	5
Arthropoda	Insect	900,000	0	<1	3	0
	Spider	35,000	0	<1	0	0
	Crustacea	50,000	0	0	<1	0
Mollusca	Gastropod	35,000	0	0	<1	0
	Cephalopod	800	0	<1	0	0
Echinoderm		7000	0	0	0	0
Annelid		9000	0	0	0	0

^1^ JAAWS mean Journal of Applied Animal Welfare Science; ^2^ AAWS means Applied Animal Welfare Science.

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
