# Peer review of "Ethics and Care: For Animals, Not Just Mammals"

_animals, 2019, doi:10.3390/ani9121018_

Round 1

Reviewer 1 Report

This manuscript reviews and analyses the human attitudes towards establishing the animal welfare and animal conservation concerns and consequent acts. It points out the anthropocentric approach that both the public and the scientists adopt, and that empathy towards certain species is the main driver defining our animal ethics conduct.

It is a very well-written manuscript with excellent flow. It gives strong arguments about how biased our attitudes are towards animals and how favoritism towards certain species functions. It is a very interesting approach, and in my opinion it definitely deserves of publication. 

Only minor improvements I can suggest: 

It would be nice if in the introduction one or two phrases are added to point out the novelty in this approach, or the difference in relation to other literature dealing with animal ethics and motives; to somehow summarize the current approaches in this subject. The Table 1 should be re-formatted accordingly to be more viewer-friendly.

Lines 312 and onwards: The discussion about fish welfare has led into a heated debate on whether they actually feel pain and are worth of ethical consideration. The author explains this situation (heated debate) largely as the result of consequences if activities dealing with fish have to be regulated. He argues that a regulatory Armageddon will be launched in this case. I really see no difference with the respective mechanisms (regulatory processes) dealing with mammals or birds raised as human foods, pets or preys. Besides there are already welfare guidelines for humane treatment and slaughtering procedures for almost all major farmed fish, dealing with each species separately (see for instance European Food Safety Authority-EFSA guidelines). In these aspects, my general impression is that the “fear” of regulatory consequences is some kind of convenient explanation, rather than depicting reality. To me it seems rather an issue of “where we draw the line” for our ethical concerns as we go lower in the phylogenetic tree, that feeds all this debate (quite similarly to the invertebrates case). I would encourage some reconsideration / restructuring of this argument. 

Author Response

I added a phrase on line 35.  

I pointed out that Browman et al's diatribe has not in fact proved correct--there is not a 'regulatory Armageddon' for fish welfare, and I agree that it is fear-mongering.  But it is not a case of 'drawing the line' between caring and neglecting, I have pointed out in a couple of places  that it is fitting the care to the species or group.

Reviewer 2 Report

This article addresses the bias of attention to only some animal species (predominantly mammals) in the general public as well as research. The author argues that many species are overlooked by the general public as well as researchers, and that this should be changed. The author gives a few suggestions how this best can be done. This paper is clearly written and a welcome and important contribution to the literature. I have a few minor comments, though, which may help to strenghten the article. 

Introduction: the introduction of the paper is very short. I'd suggest that the author already shortly presents her main thesis in the introduction, as well as the structure of the paper. As it is, the introduction is not as informative and useful as it could be.  Premise: an underlying assumption or premise of the paper is that we should care equitably for animal species (see, for example, line 74; 195), a point I agree with. However, while this point is important, it is not very much developed. I suggest that the author elaborates more why equal care or attention is important. Also, I wonder: does it really need to be equal care? Would considerably more attention to overlooked animal species not be enough? I think the paper would benefit if this underlying assumption was fleshed out more in detail.  Structure: The structure of the paper was often not very clear to me. For example, the chapter 'Popular and unpopular animals' reads as a summary of different studies, without any red thread throughout the sections. The main theses and aims of each sections could, in my opinion, be made clearer, as well as the general structure. As stated above, it would also be useful if the author made the structure of the paper in the introduction clearer. And more 'guiding flags' throughout the article would be useful, so that the reader can orient him- or herself better.  Table 1: in my version of the manuscript, there was an issue with table 1 - the different tabs and rows were not on the same level. But maybe this is only the case for the reviewer version. If not, it needs fixing before publication, so that the table is easier to read.  Chapter 'welfare of invertebrates': I found that there were quite a few repetitions between lines 353 and 399. Maybe this can be improved.  Solutions (a minor point): the author makes a few suggestions throughout the paper on how the situation for overlooked animals can be improved and how biases can be reduced. I wondered whether the decisions of funding agencies also play a role in this? Maybe it is easier to obtain funding to conduct research on 'attractive' mammals, which may also explain why there is such a big gap in the literature regarding certain species? Is there some research on this? Can author say something on this?  Conclusion: the paper does not have a conclusion section. The last paragraph summarises a bit the main theses of the article. I suggest that the author adds a real conclusion section, which summarises the results of the article as well as possible solutions to the problem presented. 

Author Response

The Introduction has been enlarged a bit, and in a couple of places I have pointed out that it needs to be appropriate care (though I did say equitable, not equal).

To make the sections more parallel, I have put Attention into each of the titles to provide a thread. 

I have pruned some of the sentences between lines 353 and 370. 

I don't think there is any research on why funding agencies make their decisions, apart from following what they know most about.

There really was a Conclusion, so I have added a header for it and inserted that researchers must pay attention to 'lower' animals, and that cephalopods may be the first group to benefit from this (I have a paper in submission detailing this, but can't cite it as it is not published--but I can foreshadow).